# Availability, affordability and stock-outs of commodities for the treatment of snakebite in Kenya

**Gaby Isabelle Ooms**[1,2]*, **Janneke van Oirschot**[1], **Dorothy Okemo**[3],
**Benjamin Waldmann**[1], **Eugene Erulu**[4], **Aukje K Mantel-Teeuwisse**[2], **Hendrika A van den Ham**[2], **Tim Reed**[1]

**1** Health Action International, Amsterdam, The Netherlands, **2** Utrecht Centre for Pharmaceutical Policy and Regulation, Division of Pharmacoepidemiology and Clinical Pharmacology, Utrecht Institute for Pharmaceutical Sciences (UIPS), Utrecht University, Utrecht, The Netherlands, **3** Access to Medicines Platform Kenya, Nairobi, Kenya, **4** Watamu Hospital, Watamu, Kenya

* gaby@haiweb.org

**Data Availability Statement:** All relevant data are within the manuscript and its Supporting information files.

## Abstract

### Background

Annually, about 2.7 million snakebite envenomings occur globally. Alongside antivenom, patients usually require additional care to treat envenoming symptoms and antivenom side effects. Efforts are underway to improve snakebite care, but evidence from the ground to inform this is scarce. This study, therefore, investigated the availability, affordability, and stock-outs of antivenom and commodities for supportive snakebite care in health facilities across Kenya.

### Methodology/principal findings

This study used an adaptation of the standardised World Health Organization (WHO)/Health Action International methodology. Data on commodity availability, prices and stock-outs were collected in July-August 2020 from public (n = 85), private (n = 36), and private not-for-profit (n = 12) facilities in Kenya. Stock-outs were measured retrospectively for a twelve-month period, enabling a comparison of a pre-COVID-19 period to stock-outs during COVID-19. Affordability was calculated using the wage of a lowest-paid government worker (LPGW) and the impoverishment approach. Accessibility was assessed combining the WHO availability target ($\geq$80%) and LPGW affordability (<1 day's wage) measures. Overall availability of snakebite commodities was low (43.0%). Antivenom was available at 44.7% of public- and 19.4% of private facilities. Stock-outs of any snakebite commodity were common in the public- (18.6%) and private (11.7%) sectors, and had worsened during COVID-19 (10.6% versus 17.0% public sector, 8.4% versus 11.7% private sector). Affordability was not an issue in the public sector, while in the private sector the median cost of one vial of antivenom was 14.4 days' wage for an LPGW. Five commodities in the public sector and two in the private sector were deemed accessible.

**Funding:** GO, JVO, DO, BW and TR were financially supported by the Lillian Lincoln Foundation and the Hennecke Foundation. The funders had no role in study design, data collection and analysis, decision to publish, or preparation of the manuscript.

**Competing interests:** The authors have declared that no competing interests exist.

**Abbreviations:** 20WBCT, 20-minute Whole Blood Clotting Test; COVID-19, Coronavirus Disease 2019; EML, Essential Medicines List; HAI, Health Action International; HHFCE, Household Final Consumption Expenditure; KSH, Kenyan Shillings; LPGW, Lowest-Paid Government Worker; NTD, Neglected Tropical Disease; PNFP, Private Not-For-Profit; UHC, Universal Health Coverage; WHO, World Health Organization.

## Conclusions

Access to snakebite care is problematic in Kenya and seemed to have worsened during COVID-19. To improve access, efforts should focus on ensuring availability at both lower- and higher-level facilities, and improving the supply chain to reduce stock-outs. Including antivenom into Universal Health Coverage benefits packages would further facilitate accessibility.

### Author summary

Annually, about 2.7 million snakebite envenomings occur globally. Treatment requires the use of antivenom, and often additional care to treat envenoming symptoms or side effects of antivenom use is also needed. Considering snakebite is common in Kenya but evidence from the ground is scarce, we set out to collect data on the availability, affordability, and stock-outs of antivenom and other commodities commonly used in the treatment of snakebite. We found that overall availability of snakebite commodities was low in Kenya, with antivenom available at 45% of public facilities, and less than 20% of private facilities. The snakebite commodities were also commonly stocked out, which seemed to have worsened during the COVID-19 pandemic. Affordability was not a problem in the public sector, where commodities were free for the patient. In the private sector, especially antivenom was unaffordable. These results highlight that the biggest issue with access to snakebite care in both the public and private sectors of Kenya was related to availability of commodities. Efforts to improve access to snakebite care in Kenya should focus on ensuring availability of the commodities at both lower- and higher-level facilities, and improving the supply chain to reduce stock-outs.

## Introduction

Snakebite has been recognised by the World Health Organization (WHO) as a neglected tropical disease (NTD) that seriously impacts people living in rural areas in Africa, Asia, Central and South America, and Oceania. It is estimated that each year, about 2.7 million snakebite envenomings occur [1]. Envenomation takes place following the bite of a venomous snake, when a mixture of toxins (venom) is injected during the bite, and can only be effectively treated with high-quality antivenom [1]. Antivenom is therefore listed on the WHO Model Essential Medicines List (EML) of priority medicines that at a minimum ought to be available in every basic healthcare system [2]. In addition to antivenom, patients usually require further care to treat the symptoms of envenoming and side effects of antivenom administration, such as anaphylactoid reactions and serum sickness [3]. Supportive care can include inter alia adrenaline, tetanus vaccine, antibiotics, airway support, intravenous fluids, pain management, blood transfusions, and assisted ventilation [3,4].

Unfortunately, in many countries antivenom is not regularly available, and sub-Saharan Africa in particular has been facing an antivenom supply crisis for at least the last 20 years [5,6]. Multiple factors contribute to this, including the limited financial resources available to sub-Saharan African countries for procurement and quality-assurance, market disincentives for manufacturers, and high dependency on antivenom imports, which have been described previously as interacting in a vicious cycle [6–11]. Also, governments generally do not

prioritise snakebite, which is also reflected in insufficient funding allocated to snakebite. For example, in 2017 the Nigerian government allocated USD 192,000 (USD 980 per million population) to its snakebite programme, which is estimated to treat only 4% of all snakebite patients [6].

Efforts to tackle this crisis are underway at the international level, with the WHO's strategy "Snakebite envenoming: A strategy for prevention and control" specifically focusing on this problem through four overarching objectives: empower and engage communities; ensure safe, effective treatment; strengthen health systems, and; increase partnerships, coordination and resources [3]. To realise these objectives, evidence from the ground is crucial [3]. Most studies on antivenom availability have been estimating the availability compared to the needs; it is estimated that the number of effective treatments available in sub-Saharan Africa may be as low as 2.5% of what is needed [7]. However, to date, in very few countries in sub-Saharan Africa has the availability of antivenoms in health facilities been methodically studied, while the availability of supportive treatment has rarely been studied in any country globally [12–14]. Further, studies on antivenom costs in sub-Saharan Africa primarily focus on wholesale prices, not on patient (out-of-pocket) prices or patient affordability [7,13]. The aim of this study was therefore to determine the availability, affordability, and stock-outs of antivenom and commodities used for supportive snakebite treatment in health facilities across Kenya to build the evidence needed to take targeted action to reduce the burden of snakebite.

Of note is that this research was undertaken in 2020, the year in which coronavirus disease 2019 (COVID-19) had an unprecedented impact on the world. Countries and their health systems were severely affected, exposing weaknesses in health systems across the globe [15]. One of the consequences of the COVID-19 pandemic has been the disruption in the manufacturing and supply of commodities [16,17]. This research provided a unique and timely opportunity to study stock-outs of snakebite commodities in Kenya during the first few months of the COVID-19 pandemic.

## Methods

### Ethics statement

This study was approved by the Amref Health Africa Ethics and Scientific Review Committee (P816/2020) and the National Commission for Science, Technology and Innovation (NACOSTI/P/20/5492). Also, letters of endorsement were obtained from the County Directors of Health of the respective counties.

### Study design and sampling

This study adopted a quantitative cross-sectional survey design with a retrospective component, using an adapted version of the standardised, gold-standard WHO/Health Action International (HAI) methodology measuring the availability, stock-outs and affordability of commodities [18]. Per this methodology, in six survey regions, 24 health facilities were randomly selected from the public, private, and private not-for-profit (PNFP) sectors, in both urban and rural locations, to function as a representative sample. A rural location was defined as an area with a population of less than 2,000 people [19]. This sampling strategy has been validated in many countries [18,20]. The six survey regions in this study were purposively sampled: four were highly snakebite endemic and HAI programme counties (Kajiado County, Kilifi County, Kwale County and Taita Taveta County), and two were less endemic (Kirinyaga County and Nyandarua County). Using the Kenya Master Health Facility List, in each county, the main public hospital was selected for inclusion, after which the other 23 licensed facilities were randomly selected under the prerequisite that they were within about an hour's drive

from the main public hospital [21]. The master list used consisted of 52 facilities in Kajiado-, 26 in Kilifi-, 48 in Kirinyaga-, 40 in Kwale-, 33 in Nyandarua- and 26 in Taita Taveta County. The selected facilities were categorised according to sector and location. The levels of health facilities surveyed ranged from level 2: dispensaries and clinics to level 6: tertiary hospitals, thereby only excluding level 1: community health services, which are not expected to stock most of the commodities surveyed in this study.

### Data collection tool

A mobile data collection application, KoBoCollect, was used to collect information about the availability, stock-outs and prices of 45 different snakebite treatment commodities. They included antivenoms, prophylactics, medicines for pain management and anaesthesia, medicines to treat complications, and several instruments and tests. Commodities were selected based on the WHO's "Guidelines for the Prevention and Clinical Management of Snakebite in Africa" [4], the "Guidelines for Prevention, Diagnosis and Management of Snakebite Envenoming in Kenya" [22], the Kenya EML 2019 [23], and consultations with clinicians and recognised snakebite experts [24]. For a full list of surveyed commodities, including their formulations and use, see S1 Table.

### Data collection

Data collectors received a one-day training, collected data in pairs, and were supervised by one of the authors (DO). They visited each of the health facilities, where a licenced healthcare worker employed at the facility assisted data collection. The presence of each commodity and formulation was recorded. Availability was defined as the presence of a survey medicine in pre-specified dose and formulation at the time of the data collection in the health facility. Patient prices were noted in Kenyan Shillings (KSH). If multiple brands of the same commodity were available, the one with the lowest patient price was taken as reference. Stock information was collected only when health facilities recorded this information in a stock-taking database, and this could be physically seen by the data collectors. Data were collected from July 28 to August 19, 2020.

### Data analysis

Data were downloaded from the server and analysed in Microsoft Excel. Data were checked for errors and outliers by the researchers (GO and DO) and double-checked with the data collectors if inconsistencies were noted. Simple descriptive statistics were used to describe the availability and affordability of commodities, and results were categorised according to sector (public, private or PNFP), and location (urban or rural).

To determine the average availability of a commodity, only health facilities that were of the level at which a specific commodity was supposed to be available as per the Kenya EML 2019 (see S1 Table), were included in the calculations [23]. An availability of 80% or higher was used as the benchmark for accessibility as per WHO guidance [25]. The combined availability of commodities that were surveyed for multiple formulations, such as amoxicillin, was calculated to provide the overall availability of that specific commodity at the facility.

Stock-outs were measured retrospectively over a twelve-month period, from 1 August 2019 to 31 July 2020. A commodity was considered stocked out if the facility usually stocked the commodity, but the stock-taking database indicated it had been out of stock at times in the past year. Stock-out information was asked for all commodities supposed to be available at that level of care, regardless of whether they were in or out of stock at the time of the survey. Taking into consideration the COVID-19 pandemic and its possible effect on the supply of

commodities, stock-out data were collected for two time periods: from 1 August 2019 to 31 January 2020, and from 1 February 2020 to 31 July 2020. Stock-outs were only calculated for commodities that had stock information available at a minimum of ten facilities per sector. Stock-outs were calculated as the percentage of facilities that reported at least one stock-out of the selected commodity over the measured time period, with stock-out days calculated as the average number of days stock-outs of a commodity lasted per facility.

Two-sample F-tests for variance and two-sample t-tests, paired t-tests, Fisher's Exact tests or binomial tests assessed whether significant differences in availability and stock-outs between and within the sectors, and between the two different time periods existed, using a significance cut-off value of 0.05.

Unit prices were calculated by dividing pack price by pack size. To calculate the affordability, two approaches were used. First, the median price of the starting dosage or full treatment course of a commodity was compared to the official salary of the lowest-paid-government worker (LPGW), which was 452.40 Kenyan Shillings (KSH) per day in 2020 [26]. If a commodity's price exceeded one day of wages, it was considered unaffordable [18]. Second, since the LPGW measure knows some limitations with representativeness as the wage of an LPGW is much higher than the income of a large proportion of the population, the impoverishment approach as developed by Van Doorslaer et al. (2006) was also used [27,28]. In this approach the impoverishing effect of purchasing a medicine is calculated by comparing the proportion of a population that is pushed below a poverty line after purchasing a medicine with the population that was already living below the poverty line [27]. The international poverty line (IPL) of USD 1.90 per person per day was used [29]. As income indicator, we used the household final consumption expenditure (HHFCE), income share per population quintile data and population size of Kenya to calculate HHFCE per capita [30]. The impoverishing effect of buying a commodity was compared to the monthly HHFCE.

Accessibility was calculated using the availability and LPGW affordability measures, resulting in a composite measure in which accessibility was achieved with an 80% or higher availability and a price of less than a day's wage for an LPGW.

## COVID-19 precautions

Data collectors took all necessary precautions as advised by the Kenyan Ministry of Health to limit the risk of COVID-19 transmission, including keeping 1.5 metres distance, wearing face masks and distributing them to participating healthcare workers, and using hand sanitiser.

## Results

### Sample

One hundred forty-four health facilities were approached to participate in the study, of which data was collected from a total of 133 health facilities from Kajiado (n = 22), Kilifi (n = 24), Kirinyaga (n = 21), Kwale (n = 24), Nyandarua (n = 20) and Taita Taveta (n = 22) counties (participation rate 92.4%). An overview of the sample characteristics is provided in Table 1. Due to the low number of facilities surveyed from the PNFP sector (n = 12), PNFP facilities were only included in the totals and were not analysed as a distinct sector.

### Availability

Availability of all surveyed commodities can be found in Table 2. Overall mean availability of the surveyed snakebite commodities in Kenya was 43.0%. No significant differences in overall mean availability between location or sector existed. Antivenom was available in 44.7% of

**Table 1. Number of surveyed facilities with availability, price and stock information available, by sector, location and level of care.**

| | Public | Private | PNFP | Total |
|---|---|---|---|---|
| Availability and price information | | | | |
| Total | 85 | 36 | 12 | 133 |
| Location | | | | |
| Urban | 26 | 22 | 8 | 56 |
| Rural | 59 | 14 | 4 | 77 |
| Level of care | | | | |
| Dispensary/clinic | 13 | 7 | 2 | 22 |
| Health centre | 53 | 18 | 5 | 76 |
| Primary hospital | 11 | 3 | 2 | 16 |
| Secondary care hospital | 4 | 4 | 1 | 9 |
| Tertiary hospital | 4 | 4 | 2 | 10 |
| Stock information | | | | |
| Total | 78 | 33 | 10 | 121 |
| Location | | | | |
| Urban | 23 | 20 | 6 | 49 |
| Rural | 55 | 13 | 4 | 72 |
| Level of care | | | | |
| Dispensary/clinic | 12 | 6 | 2 | 20 |
| Health centre | 50 | 17 | 5 | 72 |
| Primary hospital | 8 | 3 | 1 | 11 |
| Secondary care hospital | 4 | 4 | 0 | 9 |
| Tertiary hospital | 4 | 3 | 2 | 9 |

PNFP: Private not-for-profit.

public facilities, and in 19.4% of private facilities (p = 0.009). Availability differed significantly between urban and rural locations within the public sector (p = 0.003). None of the level 2 facilities stocked antivenom, while more than 70% of level 4 and 5 public facilities (primary and secondary hospitals) did stock antivenom (see Table 3). Availability of both antivenom and adrenaline, which should be available in case of anaphylaxis as a consequence of anti-venom usage, was lower: 36.4% and 25.0% of level 4 and 5 facilities, respectively, had both available. Availability of antivenom in highly endemic counties was 41.8%, availability in less endemic counties was 19.0% (p = 0.01). The most commonly stocked antivenom brands in the public sector were Snake Venom Antiserum (African IHS) by VINS Bioproducts Ltd (66.7% of facilities), and Inoserp PAN-AFRICAN by INOSAN Biopharma (33.3% of facilities) (see S2 Table).

In general, antibiotics had a relatively high availability of 46.8% to 91.0%. Availability of commodities used for the management of complications was more variable, with hydrocorti-sone having the highest availability (79.7%). Significant differences in availability existed for adrenaline, chlorpheniramine, and prednisolone. Paracetamol had the highest availability of commodities used for pain management. Blood products had a very low availability across sectors, and variability in availability of medical instruments and materials was observed.

## Stock-outs

Stock information was available for 121 of 133 facilities (91.0%, see Table 1). Overall, on average 18.6% of all public facilities reported at least one stock-out of any of the surveyed

**Table 2. Availability of snakebite commodities in Kenya, per sector and location.**

| Commodities | Overall[a] | | | | Public | | | | Private | | | | |
|---|---|---|---|---|---|---|---|---|---|---|---|---|---|
| | Urban | Rural | p-value | Total | Urban | Rural | p-value | Total | Urban | Rural | p-value | Total | p-value[h] |
| **Total** | 44.5 | 42.2 | 0.751 | 44.0 | 45.0 | 47.8 | 0.708 | 43.4 | 45.1 | 47.2 | 0.773 | 46.3 | 0.700 |
| **Antivenom and anti-tetanus** | | | | | | | | | | | | | |
| Antivenom | 44.6 | 28.6 | 0.056 | 35.3 | 69.2 | 33.9 | **0.003** | 44.7 | 27.3 | 7.1 | 0.078 | 19.4 | **0.009** |
| Tetanus vaccine | 66.1 | 79.2 | 0.089 | 73.7 | 53.8 | 74.6 | 0.059 | 68.2 | 77.3 | 92.9 | 0.169 | 83.3 | 0.088 |
| **Antibiotics** | | | | | | | | | | | | | |
| Benzylpenicillin | 58.9 | 64.9 | 0.480 | 62.4 | 61.5 | 64.4 | 0.800 | 63.5 | 72.7 | 64.3 | 0.592 | 69.4 | 0.053 |
| Metronidazole[b] | 87.5 | 93.5 | 0.233 | 91.0 | 88.5 | 96.6 | 0.141 | 94.1 | 86.4 | 85.7 | 0.956 | 86.1 | 0.144 |
| Gentamicin[c] | 69.6 | 68.8 | 0.920 | 69.2 | 73.1 | 71.2 | 0.858 | 71.8 | 72.7 | 57.1 | 0.245 | 66.7 | 0.575 |
| Amoxicillin[d] | 94.6 | 93.5 | 0.786 | 94.0 | 96.2 | 94.9 | 0.804 | 95.3 | 90.9 | 85.7 | 0.629 | 88.9 | 0.195 |
| Amoxicillin + clavulanic acid[f] | 53.1 | 41.9 | 0.243 | 46.8 | 62.5 | 45.8 | 0.355 | 51.4 | 44.4 | 27.3 | 0.182 | 37.9 | 0.220 |
| **Complications management** | | | | | | | | | | | | | |
| Adrenaline | 60.7 | 41.6 | **0.029** | 49.6 | 61.5 | 35.6 | **0.026** | 43.5 | 54.5 | 71.4 | 0.311 | 61.1 | 0.077 |
| Hydrocortisone | 82.1 | 77.9 | 0.550 | 79.7 | 80.8 | 79.7 | 0.906 | 80.0 | 81.8 | 71.4 | 0.645 | 77.8 | 0.783 |
| Chlorpheniramine[e,f] | 40.8 | 11.3 | **<0.001** | 24.3 | 29.2 | 4.2 | **0.002** | 12.5 | 61.1 | 27.3 | 0.077 | 48.3 | **<0.001** |
| Prednisolone[g] | 73.2 | 45.5 | **0.002** | 57.1 | 53.8 | 32.2 | 0.060 | 38.8 | 86.4 | 85.7 | 0.592 | 86.1 | **<0.001** |
| Neostigmine[g] | 18.2 | 0.0 | 0.508 | 17.1 | 21.1 | NA | NA | 21.1 | 22.2 | 0.0 | 0.461 | 18.2 | 0.850 |
| Atropine[f] | 55.1 | 54.8 | 0.978 | 55.0 | 50.0 | 54.2 | 0.738 | 52.8 | 66.7 | 45.5 | 0.260 | 58.6 | 0.594 |
| **Pain management** | | | | | | | | | | | | | |
| Paracetamol | 96.4 | 88.3 | 0.093 | 91.7 | 96.2 | 84.7 | 0.133 | 88.2 | 95.5 | 100.0 | 0.418 | 97.2 | 0.116 |
| Dihydrocodeine phosphate[f] | 10.2 | 0.0 | **0.010** | 4.5 | 12.5 | 0.0 | **0.012** | 4.2 | 5.6 | 0.0 | 0.426 | 3.4 | 0.867 |
| Morphine[g] | 15.2 | 0.0 | 0.552 | 14.3 | 15.8 | NA | NA | 15.8 | 22.2 | 0.0 | 0.461 | 18.2 | 0.865 |
| **Local anaesthesia** | | | | | | | | | | | | | |
| Lidocaine | 71.4 | 76.6 | 0.498 | 74.4 | 73.1 | 76.3 | 0.753 | 75.3 | 77.3 | 71.4 | 0.693 | 75.0 | 0.973 |
| **Fluids** | | | | | | | | | | | | | |
| Saline | 66.1 | 79.2 | 0.089 | 73.7 | 76.9 | 83.1 | 0.505 | 81.2 | 54.5 | 78.6 | 0.143 | 63.9 | **0.042** |
| Fresh frozen plasma[g] | 6.1 | 0.0 | 0.720 | 5.7 | 5.3 | NA | NA | 5.3 | 0.0 | 0.0 | NA | 0.0 | 0.439 |
| Blood platelets[g] | 3.0 | 0.0 | 0.803 | 2.9 | 5.3 | NA | NA | 5.3 | 0.0 | 0.0 | NA | 0.0 | 0.439 |
| Red blood cells[g] | 9.1 | 0.0 | 0.656 | 8.6 | 5.3 | NA | NA | 5.3 | 11.1 | 0.0 | 0.621 | 9.1 | 0.685 |
| Whole blood[g] | 15.2 | 0.0 | 0.552 | 14.3 | 15.8 | NA | NA | 15.8 | 11.1 | 0.0 | 0.621 | 9.1 | 0.603 |
| **Medical instruments, materials** | | | | | | | | | | | | | |
| Bandage | 66.1 | 85.7 | **0.007** | 77.4 | 61.5 | 88.1 | **0.005** | 80.0 | 72.7 | 85.7 | **0.004** | 77.8 | 0.783 |
| Splint[f] | 0.0 | 6.5 | 0.070 | 3.6 | 0.0 | 4.2 | 0.310 | 2.8 | 0.0 | 18.2 | 0.061 | 6.9 | 0.337 |
| Sticking plaster[f] | 26.5 | 27.4 | 0.917 | 27.0 | 29.2 | 25.0 | 0.705 | 26.4 | 27.8 | 45.5 | 0.331 | 34.5 | 0.416 |
| Oxygen cylinder | 44.6 | 41.6 | 0.723 | 42.9 | 53.8 | 37.3 | 0.155 | 42.4 | 36.4 | 64.3 | 0.102 | 47.2 | 0.622 |
| Laryngoscope | 7.1 | 5.2 | 0.641 | 6.0 | 3.8 | 1.7 | 0.547 | 2.4 | 13.6 | 14.3 | 0.956 | 13.9 | **0.013** |
| Cuffed endotracheal tube[f] | 12.2 | 16.1 | 0.563 | 14.4 | 16.7 | 12.5 | 0.630 | 13.9 | 5.6 | 36.4 | **0.033** | 17.2 | 0.668 |
| Nasal prong | 41.1 | 49.4 | 0.344 | 45.9 | 38.5 | 45.8 | 0.532 | 43.5 | 45.5 | 64.3 | 0.270 | 52.8 | 0.351 |
| Ambu bag | 55.4 | 63.6 | 0.336 | 60.2 | 69.2 | 69.5 | 0.981 | 69.4 | 45.5 | 42.9 | 0.878 | 44.4 | **0.010** |
| Oral airway[f] | 18.4 | 25.8 | 0.352 | 22.5 | 12.5 | 20.8 | 0.386 | 18.1 | 27.8 | 54.5 | 0.149 | 37.9 | **0.034** |
| Ventilator[f] | 14.3 | 3.2 | **0.034** | 8.1 | 8.3 | 4.2 | 0.467 | 5.6 | 16.7 | 0.0 | 0.153 | 10.3 | 0.391 |
| Intravenous cannula | 67.9 | 80.5 | 0.095 | 75.2 | 73.1 | 78.0 | 0.624 | 76.5 | 63.6 | 92.9 | **0.048** | 75.0 | 0.862 |
| Catheter | 64.3 | 66.2 | 0.816 | 65.4 | 73.1 | 66.1 | 0.524 | 68.2 | 50.0 | 64.3 | 0.400 | 55.6 | 0.183 |
| Syringe + needle | 92.9 | 98.7 | 0.080 | 96.2 | 92.3 | 98.3 | 0.167 | 96.5 | 95.5 | 100.0 | 0.418 | 97.2 | 0.833 |
| IV administration set | 71.4 | 85.7 | **0.043** | 79.7 | 76.9 | 84.7 | 0.383 | 82.4 | 68.2 | 92.9 | 0.083 | 77.8 | 0.558 |
| Urine dipstick | 57.4 | 58.4 | 0.210 | 57.9 | 57.7 | 57.6 | 0.212 | 57.6 | 59.1 | 57.1 | 0.650 | 58.3 | 0.944 |

*(Continued)*

**Table 2.** (Continued)

| Commodities | Mean Availability (%) | | | | | | | | | | | | |
|---|---|---|---|---|---|---|---|---|---|---|---|---|---|
| | Overall[a] | | | | Public | | | | Private | | | | |
| | Urban | Rural | p-value | Total | Urban | Rural | p-value | Total | Urban | Rural | p-value | Total | p-value[h] |
| Creatinine clearance blood test | 14.3 | 5.2 | 0.071 | 9.0 | 11.5 | 0.0 | **0.008** | 3.5 | 13.6 | 28.6 | 0.270 | 19.4 | **0.004** |
| Blood urea nitrogen testing | 10.7 | 5.2 | 0.233 | 7.5 | 3.8 | 1.7 | 0.547 | 2.4 | 13.6 | 21.4 | 0.541 | 16.7 | **0.004** |
| 20WBCT | 10.7 | 5.2 | 0.233 | 7.5 | 11.5 | 0.0 | **0.008** | 3.5 | 13.6 | 21.4 | 0.541 | 16.7 | **0.012** |
| Point-of-Care INR device | 5.4 | 1.3 | 0.176 | 3.0 | 0.0 | 0.0 | NA | 0.0 | 9.1 | 7.1 | 0.837 | 8.3 | **0.007** |

20WBCT: 20-minute whole blood clotting test; IV: Intravenous; INR: International normalised ratio; NA: Commodity not surveyed because no facility was included that ought to have the commodity available.

[a]Availability includes the private not-for-profit sector.

[b]Metronidazole combines the availability of metronidazole 200mg, 400mg and 200mg/5ml.

[c]Gentamicin combines the availability of gentamicin 10mg/2ml, 20mg/2ml, 40mg/2ml and 80mg/2ml.

[d]Amoxicillin combines the availability of amoxicillin 250mg and 500mg.

[e]Chlorpheniramine combines the availability of chlorpheniramine 10mg/1ml and 2mg/5ml.

[f]Available from the health centre level and up.

[g]Available from the primary hospital level and up.

[h]Level of significance between public and private sector.

Availability:

White: <20%; Very light gray: 20–39.9%; Light gray: 40–59.9%; Gray: 60–79.9%; Dark gray: ≥80%


**Table 3. Availability of antivenom, and antivenom and adrenaline, per level and sector.**

| | Antivenom availability (%) | | | | | | | | |
|---|---|---|---|---|---|---|---|---|---|
| | Overall[a] | | | Public | | | Private | | |
| | Urban | Rural | Total | Urban | Rural | Total | Urban | Rural | Total |
| **Level 2** | 0.0 | 0.0 | 0.0 | 0.0 | 0.0 | 0.0 | 0.0 | 0.0 | 0.0 |
| **Level 3** | 37.5 | 36.7 | 36.8 | 60.0 | 41.7 | 43.4 | 33.3 | 11.1 | 22.2 |
| **Level 4** | 47.4 | 0.0 | 45.0 | 72.7 | NA | 72.7 | 25.0 | 0.0 | 20.0 |
| **Level 5** | 62.5 | 0.0 | 55.6 | 75.0 | NA | 75.0 | 33.3 | 0.0 | 25.0 |
| **Level 6** | 83.3 | NA | 83.3 | 100.0 | NA | 100.0 | 50.0 | NA | 50.0 |
| | **Antivenom and adrenaline availability (%)** | | | | | | | | |
| | Overall[a] | | | Public | | | Private | | |
| | Urban | Rural | Total | Urban | Rural | Total | Urban | Rural | Total |
| **Level 2** | 0.0 | 0.0 | 0.0 | 0.0 | 0.0 | 0.0 | 0.0 | 0.0 | 0.0 |
| **Level 3** | 25.0 | 18.3 | 19.7 | 40.0 | 20.8 | 22.6 | 22.2 | 0.0 | 11.1 |
| **Level 4** | 26.3 | 0.0 | 25.0 | 36.4 | NA | 36.4 | 25.0 | 0.0 | 20.0 |
| **Level 5** | 37.5 | 0.0 | 33.3 | 25.0 | NA | 25.0 | 33.3 | 0.0 | 25.0 |
| **Level 6** | 83.3 | NA | 83.3 | 100.0 | NA | 100.0 | 50.0 | NA | 50.0 |

NA: Not applicable.

Level 2: Dispensary/clinic; level 3: Health centre; level 4: Primary hospital; level 5: Secondary hospital; level 6: Tertiary hospital.

[a]Includes the private not-for-profit sector.

Availability:

White: <20%; Very light gray: 20–39.9%; Light gray: 40–59.9%; Gray: 60–79.9%; Dark gray: ≥80%

commodities over a twelve-month period, with stock-outs lasting on average 30.5 days per facility (see Table 4). In the private sector, stock-outs occurred on average in 11.7% of the facilities over the twelve-month period and lasted on average 24.0 days per facility. In both sectors stock-outs of almost all commodities occurred significantly more often from February to July 2020 than from August 2019 to January 2020.

Over a twelve-month period, 20.0% of all public facilities experienced a stock-out of antivenom, averaging 13.6 days per facility. No data on antivenom stock-outs in the private sector was available due to the small sample of health facilities with stock information for antivenom. Duration of stock-outs was longest for oxygen cylinders, hydrocortisone and chlorpheniramine (10mg/1ml) in the public sector, and for metronidazole and tetanus vaccine in the private sector.

## Affordability

Pricing information was not provided for 11.2% (110/979) and 32.4% (145/303) of available commodities in the public and private sectors, respectively. Using the wage of an LPGW, in the public sector all commodities were affordable to the patient; none of the commodities cost more than a day's wage if the median price was considered the benchmark (see Table 5). However, when looking at the maximum price paid for the commodities at public facilities, one vial of antivenom can cost up to 44.2 days of wages. In the private sector, four commodities were unaffordable for an LPGW, with the median cost of one vial of antivenom being 14.4 days of wages. Benzylpenicillin, gentamicin (10mg or 20mg/2ml), and morphine were also unaffordable in the private sector. Using the impoverishment approach, it was calculated that 24.2% of the population was already living below the IPL. In the public sector, purchasing any medicines at median price had a minimal impoverishing effect. In the private sector, however, purchasing one vial of antivenom at median price would push 39.0% of the population below the IPL. Other impoverishing purchases included benzylpenicillin, gentamicin, hydrocortisone and morphine. When purchasing a vial of antivenom at the maximum price at a public or private facility (KSH 20,000.00), for 63.3% of the population treatment would be unaffordable and they would be impoverished. Box 1 provides a real-life example of the affordability of treatment received by a snakebite patient with a typical disease course.

## Accessibility

In the public sector, five of 23 commodities were deemed accessible, as they cost less than a day's wage for an LPGW and were available at 80% or more of health facilities (see Fig 1 and S3 Table). These commodities were: metronidazole (200mg or 400mg), amoxicillin (250mg), paracetamol, hydrocortisone and saline. In the private sector, two of 23 commodities (paracetamol and prednisolone) were accessible. In both sectors the main problem was low availability, as 18 of 23 commodities in the public sector and 15 of 23 commodities in the private sector cost less than a day's wage for an LPGW but had an availability of below 80%.

Accessibility of antivenom was variable (see Fig 2). Antivenom was accessible (both available and affordable) in 35% of public- and in 3% of private facilities, and available but not affordable in 8% of public- and 13% of private facilities. In the remaining facilities no antivenom was available.

## Discussion

This study is the first to research the availability, stock-outs and affordability of 45 commodities used in the treatment of snakebites in Kenya. It showed that overall availability of the commodities was low (43.0%). Antivenom was available at 44.7% of public facilities and 19.4% of

**Table 4. Facilities reporting stock-outs of snakebite commodities and average number of stock-out days per facility over a six- and twelve-month period, per sector.**

| | % of facilities reporting a stock-out | | | | | | | | | Average number of stock-out days per facility | | | | | | | | |
| | Public | | | | Private | | | | | Public | | | | Private | | | | |
| | Aug-Jan[a] | Feb-July[a] | p-value | Aug-July[b] | Aug-Jan[a] | Feb-July[a] | p-value | Aug-July[b] | p-value[c] | Aug-Jan[a] | Feb-July[a] | p-value | Aug-July[b] | Aug-Jan[a] | Feb-July[a] | p-value | Aug-July[b] | p-value[c] |
|---|---|---|---|---|---|---|---|---|---|---|---|---|---|---|---|---|---|---|
| **Total** | 10.6 | 17.0 | **<0.001** | 18.6 | 8.4 | 11.7 | **0.005** | 11.7 | 0.141 | 22.9 | 16.9 | 0.457 | 30.5 | 14.3 | 13.2 | 0.861 | 24.0 | 0.589 |
| **Antivenom and anti-tetanus** | | | | | | | | | | | | | | | | | | |
| Antivenom | 20.0 | 11.4 | 0.685 | 20.0 | NA | NA | NA | NA | NA | 11.9 | 3.0 | 0.316 | 13.6 | NA | NA | NA | NA | NA |
| Tetanus vaccine | 15.0 | 30.0 | **0.039** | 35.0 | 22.7 | 22.7 | 0.382 | 27.3 | 0.509 | 46.3 | 19.3 | 0.248 | 36.4 | 27.6 | 27.2 | 0.986 | 45.7 | 0.769 |
| **Antibiotics** | | | | | | | | | | | | | | | | | | |
| Benzylpenicillin | 16.3 | 16.3 | 0.405 | 20.4 | 12.5 | 12.5 | 0.323 | 12.5 | 0.479 | 26.4 | 30.3 | 0.797 | 45.3 | 8.5 | 12.5 | 0.726 | 21.0 | 0.571 |
| Metronidazole (200mg or 400mg) | 6.9 | 6.9 | 0.372 | 8.6 | 13.0 | 17.4 | 0.171 | 17.4 | 0.257 | 4.3 | 3.0 | 0.681 | 5.8 | 3.7 | 16.8 | 0.435 | 19.5 | 0.401 |
| Metronidazole (200mg/5ml) | 0.0 | 7.7 | **<0.001** | 7.7 | 18.2 | 18.2 | 0.323 | 18.2 | 0.439 | NS | 7.0 | NA | 7.0 | 97.5 | 45.0 | 0.617 | 142.5 | NA |
| Gentamicin (10mg or 20mg/2ml) | 6.5 | 9.7 | 0.242 | 9.7 | NA | NA | NA | NA | NA | 11.0 | 3.0 | 0.347 | 10.3 | NA | NA | NA | NA | NA |
| Gentamicin (40mg or 80mg/2ml) | 5.0 | 7.5 | 0.138 | 7.5 | 18.2 | 18.2 | 0.323 | 18.2 | 0.291 | 25.0 | 6.3 | **0.030** | 23.0 | 5.5 | 5.5 | 1.00 | 11.0 | 0.491 |
| Amoxicillin (250mg) | 3.2 | 6.5 | **0.048** | 6.5 | 9.1 | 9.1 | 0.323 | 9.1 | 0.680 | 3.0 | 3.0 | 1.00 | 4.5 | 11.0 | 16.0 | 0.349 | 27.0 | **0.002** |
| Amoxicillin (500mg) | 13.6 | 47.7 | **<0.001** | 47.7 | 9.5 | 14.3 | 0.133 | 14.3 | **0.009** | 2.3 | 17.7 | **0.002** | 18.4 | 10.5 | 7.0 | 0.080 | 14.0 | 0.711 |
| Amoxicillin + clavulanic acid | 17.1 | 25.7 | 0.064 | 31.4 | NA | NA | NA | NA | NA | 22.2 | 20.3 | 0.911 | 28.7 | NA | NA | NA | NA | NA |
| **Complications management** | | | | | | | | | | | | | | | | | | |
| Adrenaline | 17.9 | 30.8 | **0.015** | 33.3 | 5.0 | 15.0 | **0.016** | 15.0 | 0.133 | 9.1 | 18.1 | 0.267 | 21.6 | 4.0 | 13.7 | 0.629 | 15.0 | 0.371 |
| Hydrocortisone | 10.0 | 13.3 | 0.142 | 13.3 | 7.7 | 15.4 | **0.046** | 15.4 | 0.801 | 37.0 | 48.1 | 0.782 | 75.9 | 4.0 | 3.5 | 0.900 | 5.5 | 0.172 |
| Chlorpheniramine (10mg/1ml) | 52.4 | 71.4 | **0.023** | 71.4 | 7.7 | 23.1 | **0.014** | 23.1 | **0.006** | 45.2 | 41.0 | 0.868 | 74.3 | 1.0 | 6.7 | 0.425 | 7.0 | **0.039** |
| Chlorpheniramine (2mg/5ml) | 33.3 | 53.3 | **0.031** | 53.3 | 14.3 | 21.4 | 0.128 | 21.4 | 0.077 | 9.0 | 8.3 | 0.909 | 13.9 | 3.0 | 3.3 | 0.913 | 5.3 | 0.248 |
| Prednisolone | 26.8 | 43.9 | **0.006** | 43.9 | 4.2 | 8.3 | 0.078 | 8.3 | **0.003** | 34.1 | 28.1 | 0.735 | 48.9 | 5.0 | 10.0 | 0.602 | 12.5 | **0.045** |
| Neostigmine | 30.0 | 40.0 | 0.150 | 40.0 | NA | NA | NA | NA | NA | 8.3 | 14.3 | 0.500 | 20.5 | NA | NA | NA | NA | NA |
| Atropine | 7.1 | 19.0 | **0.006** | 19.0 | 16.7 | 16.7 | 0.395 | 16.7 | 0.851 | 5.7 | 19.5 | 0.105 | 21.6 | 18.7 | 20.3 | 0.895 | 39.0 | 0.273 |
| **Pain management** | | | | | | | | | | | | | | | | | | |
| Paracetamol | 1.5 | 6.0 | **0.003** | 6.0 | 3.8 | 3.8 | **<0.001** | 3.8 | 0.684 | 40.0 | 38.0 | 0.965 | 48.0 | 20.0 | 15.0 | NA | 35.0 | 0.788 |
| **Local anaesthesia** | | | | | | | | | | | | | | | | | | |
| Lidocaine | 7.0 | 14.0 | **0.017** | 14.0 | 10.0 | 5.0 | 0.323 | 10.0 | 0.644 | 12.3 | 10.8 | 0.849 | 16.9 | 6.5 | 20.0 | 0.269 | 16.5 | 0.982 |
| **Fluids** | | | | | | | | | | | | | | | | | | |
| Saline | 1.6 | 4.9 | **0.017** | 4.9 | 0.0 | 5.3 | **<0.001** | 5.3 | 0.952 | 1.0 | 5.7 | NA | 6.0 | NS | 10.0 | NA | 10.0 | NA |
| **Instruments and materials** | | | | | | | | | | | | | | | | | | |
| Bandage | 0.0 | 0.0 | **<0.001** | 0.0 | 4.3 | 13.0 | **0.016** | 13.0 | **0.007** | NS | NS | NS | NS | 3.0 | 3.3 | NA | 4.3 | NA |
| Sticking plaster | 15.0 | 30.0 | **0.022** | 30.0 | NA | NA | NA | NA | NA | 12.0 | 27.5 | 0.438 | 33.5 | NA | NA | NA | NA | NA |
| Oxygen cylinder | 11.1 | 5.6 | 0.609 | 11.1 | NA | NA | NA | NA | NA | 180.0 | 60.0 | NA | 210.0 | NA | NA | NA | NA | NA |
| Nasal prong | 3.8 | 11.5 | **0.016** | 11.5 | NA | NA | NA | NA | NA | 1.0 | 1.3 | NA | 1.7 | NA | NA | NA | NA | NA |
| Ambu bag | 0.0 | 0.0 | **<0.001** | 0.0 | NA | NA | NA | NA | NA | NS | NS | NS | NS | NA | NA | NA | NA | NA |
| Intravenous cannula | 2.3 | 2.3 | **<0.001** | 2.3 | 0.0 | 0.0 | **<0.001** | 0.0 | 0.569 | 2.0 | 1.0 | NA | 3.0 | NS | NS | NS | NS | NA |
| Catheter | 0.0 | 2.4 | **<0.001** | 2.4 | 0.0 | 0.0 | **<0.001** | 0.0 | 0.601 | NS | 1.0 | NA | 1.0 | NS | NS | NS | NS | NS |

*(Continued)*

**Table 4.** (Continued)

| | % of facilities reporting a stock-out | | | | | | | | | Average number of stock-out days per facility | | | | | | | | |
| | Public | | | | Private | | | | | Public | | | | Private | | | | |
| | Aug-Jan[a] | Feb-July[a] | p-value | Aug-July[b] | Aug-Jan[a] | Feb-July[a] | p-value | Aug-July[b] | p-value[c] | Aug-Jan[a] | Feb-July[a] | p-value | Aug-July[b] | Aug-Jan[a] | Feb-July[a] | p-value | Aug-July[b] | p-value[c] |
|---|---|---|---|---|---|---|---|---|---|---|---|---|---|---|---|---|---|---|
| Syringe + needle | 0.0 | 0.0 | **<0.001** | 0.0 | 0.0 | 0.0 | **<0.001** | 0.0 | NA | NS | NS | NS | NS | NS | NS | NS | NS | NS |
| IV administration set | 0.0 | 0.0 | **<0.001** | 0.0 | 0.0 | 6.3 | **<0.001** | 6.3 | 0.067 | NS | NS | NS | NS | NS | 2.0 | NA | 2.0 | NA |
| Urine dipstick | 5.4 | 5.4 | 0.323 | 5.4 | NA | NA | NA | NA | NA | 1.0 | 1.5 | NA | 2.0 | NA | NA | NA | NA | NA |

NA: Not included due to small sample; NS: No stock-out.

[a]Stock-outs measured over a six-month period.

[b]Stock-outs measured over a twelve-month period.

[c]Level of significance between public and private sector.

private facilities. Stock-outs of commodities were common in both the public (18.6%) and private (11.7%) sectors, lasting on average about a month in the public sector and 24 days in the private sector over a twelve-month period. Stock-outs seemed to have worsened during COVID-19, with facilities reporting stock-outs significantly more often in the period of February to July 2020 than in August 2019 to January 2020. Affordability was not an issue in the public sector, as most commodities were free to the patient. In the private sector, affordability was a slightly bigger problem, especially when buying antivenom: it cost an LPGW 14.4 days of wages, and would impoverish 39.0% of the population if they required treatment. This study further showed that only five commodities in the public sector and two in the private sector could be considered accessible. The biggest issue in both sectors was availability.

Mean availability in rural public facilities was higher than in urban public facilities. One of the explanations is that rural facilities in Kenya are often lower-level facilities where more specialised commodities such as morphine and blood products are not supposed to be available as per the Kenya EML. This shows that not only availability is affected by supply chain issues such as stock-outs, but patients accessing care at lower-level facilities are facing a barrier that is inherent to the system: these more specialised commodities are never available at these levels of care. Further, antivenom availability was generally low in Kenya, and was higher for urban facilities than rural facilities. While the Kenya EML stipulates antivenom to be available at the dispensary/clinic level and up, in reality this is not the case. Since most snakebites occur in rural areas where often only lower level facilities such as health centres are found, this study confirms the discrepancy in antivenom availability which is often referred to in literature: it is most often unavailable in the places where it is most needed [31]. These findings also confirm that snakebite patients are often forced to travel greater distances to reach a health facility where antivenom is available. To improve availability, specific attention should be paid to availability at lower-level facilities, especially in the case of antivenom, where timely administration is crucial. Further, 20% of the facilities that did stock antivenom experienced stock-outs, which lasted on average 13.6 days per facility. This seems to suggest that while one-fifth of facilities experienced stock-outs of antivenom, the supply is more or less consistent, with stock-outs, although still troubling, not occurring for extended periods of time. Nevertheless, this study did not ask about the number of vials available at the facility. Since several vials are required per treatment per patient, the supply could be consistent for a period but not necessarily meet treatment demand. Focus should therefore be on making antivenom adequately available in facilities, especially where it is not yet stocked.

**Table 5. Affordability of snakebite commodities, using the wage of an LPGW and the impoverishment approach, per sector.**

| | Treatment regimen | Affordability for LPGW (days of wage)[a] | | Additional population below IPL, post-purchase (%)[b] | |
|---|---|---|---|---|---|
| | | Public | Private | Public | Private |
| **Antivenom and anti-tetanus** | | | | | |
| Antivenom | 1 vial[c] | 0.0 (0.0–44.2) | 14.4 (0.7–44.2) | 0.0 (0.0–63.3) | 39.0 (2.5–63.3) |
| Tetanus vaccine | 1 vial | 0.0 (0.0–0.2) | 0.2 (0.0–0.6) | 0.0 (0.0–0.8) | 0.8 (0.0–2.1) |
| **Antibiotics** | | | | | |
| Benzylpenicillin | 20 vials | 0.0 (0.0–4.4) | 4.4 (0.0–22.1) | 0.0 (0.0–15.1) | 15.1 (0.0–51.2) |
| Metronidazole (200mg or 400mg) | 15 tablets | 0.0 (0.0–1.0) | 0.2 (0.0–2.3) | 0.0 (0.0–3.9) | 0.6 (0.0–8.4) |
| Metronidazole (200mg/5ml) | 15 vials | 0.0 (0.0–2.1) | 0.1 (0.0–13.3) | 0.0 (0.0–7.7) | 0.0 (0.0–45.8) |
| Gentamicin (10mg or 20mg/2ml) | 5 vials | 0.0 (0.0–1.1) | 2.2 (1.7–2.2) | 0.0 (0.0–4.0) | 8.0 (6.3–8.0) |
| Gentamicin (40mg or 80mg/2ml) | 3 vials | 0.0 (0.0–0.7) | 0.6 (0.0–1.3) | 0.0 (0.0–2.5) | 2.3 (0.0–5.1) |
| Amoxicillin (250mg) | 15 tablets | 0.0 (0.0–0.2) | 0.2 (0.0–1.0) | 0.0 (0.0–0.6) | 0.8 (0.0–3.8) |
| Amoxicillin (500mg) | 15 tablets | 0.0 (0.0–2.0) | 0.3 (0.0–2.3) | 0.0 (0.0–7.3) | 1.3 (0.0–8.4) |
| Amoxicillin + clavulanic acid | 15 tablets | 0.0 (0.0–8.3) | 1.0 (0.0–1.7) | 0.0 (0.0–7.7) | 3.8 (0.0–6.3) |
| **Complications management** | | | | | |
| Adrenaline | 1 vial[c] | 0.0 (0.0–0.1) | 0.2 (0.0–0.4) | 0.0 (0.0–0.4) | 0.8 (0.0–1.7) |
| Hydrocortisone | 6 vials | 0.0 (0.0–1.3) | 1.3 (0.0–2.7) | 0.0 (0.0–5.1) | 5.1 (0.0–9.0) |
| Chlorpheniramine (10mg/1ml) | 6 vials | 0.1 (0.0–1.3) | 0.7 (0.0–2.7) | 0.5 (0.0–5.1) | 2.5 (0.0–9.0) |
| Chlorpheniramine (2mg/5ml) | 6 vials | 0.0 (0.0–4.1) | 0.3 (0.0–1.3) | 0.0 (0.0–14.3) | 1.3 (0.0–5.1) |
| Prednisolone | 20 tablets | 0.0 (0.0–0.2) | 0.1 (0.0–0.4) | 0.0 (0.0–0.8) | 0.5 (0.0–1.7) |
| Neostigmine | 1 vial[c] | 0.1 (0.0–0.2) | 0.5 (0.2–0.8) | 0.2 (0.0–0.8) | 1.8 (0.6–3.0) |
| Atropine | 1 vial[c] | 0.0 (0.0–0.1) | 0.2 (0.0–0.4) | 0.0 (0.0–0.6) | 0.1 (0.0–1.7) |
| **Pain management** | | | | | |
| Paracetamol | 18 tablets | 0.0 (0.0–0.1) | 0.1 (0.0–0.4) | 0.0 (0.0–0.3) | 0.3 (0.0–1.5) |
| Dihydrocodeine phosphate | 3 tablets[c] | 0.0 (0.0–0.0) | NP | 0.0 (0.0–0.3) | NP |
| Morphine | 1 vial[c] | 0.2 (0.0–0.3) | 1.1 (0.0–2.2) | 0.7 (0.0–1.1) | 4.2 (0.0–8.0) |
| **Local anaesthesia** | | | | | |
| Lidocaine | 1 tube | 0.0 (0.0–0.2) | 0.0 (0.0–3.3) | 0.0 (0.0–0.8) | 0.1 (0.0–11.6) |
| **Fluids** | | | | | |
| Saline | 2 litres | 0.0 (0.0–0.9) | 0.6 (0.0–6.2) | 0.0 (0.0–3.4) | 2.4 (0.0–20.8) |

IPL: international poverty line; LPGW: Lowest-paid government worker.

[a]Calculated using the median price of a medicine.

[b]IPL of USD 1.90 was equal to KSH 202.58 on August 1 2020.

[c]Starting dose.

[d]Repeat after one, six and twelve months.

This study further showed that Snake Venom Antiserum (African IHS) produced by VINS Bioproducts Ltd, and Inoserp PAN-AFRICAN produced by INOSAN Biopharma were the most commonly stocked antivenom products, which have been shown to be ineffective in pre-clinical tests for some of the most commonly found snakes in Kenya. Especially for the VINS antivenom, its use is not supported by any pre-clinical data [10]. Important to further note is that no antivenom in use in Kenya is supported by data from a randomised controlled trial. The use of ineffective antivenoms has been shown in some studies to lead to avoidable deaths. In Ghana, for instance, a switch from Sanofi's FAV-Afrique antivenom, one of the only safe and effective antivenoms previously used in Africa but discontinued due to commercial interests, to another antivenom led to an increase in mortality rate, from 1.8% to 12.1% [32].

### Box 1. Affordability of snakebite treatment for a patient

A five-year-old boy was bitten by an unidentified snake near the Tana River in Southern Kenya. As traditional treatment, a black stone was applied to the site of the bite, after which the patient was taken to the nearest dispensary (public sector). There he was given a hydrocortisone injection* and tetanus toxoid vaccine. After, the patient was referred to a general hospital (public sector) 100km away. There he was administered two vials of antivenom and referred to a private nursing home. The patient stayed 11 days at the nursing home, he was discharged due to the family's financial constraints. During the 11-day stay, the patient received two more vials of antivenom, one vial of adrenaline as premedication, amoxicillin + clavulanic acid twice daily for the entire treatment and gentamicin 40mg/2ml once daily for five days after developing bite site soft tissue sepsis, and paracetamol four times daily for six days. Final health outcome is unknown due to his return to his hometown.

The costs of treatment alone, using the median treatment costs found in this study, would amount to KSH 14,258.00. For an LPGW, this would be 31.5 days of wage, and an additional 59.0% of the population would be pushed below the IPL if they needed this treatment. If the entire treatment were provided in private health facilities, an LPGW would need to work for 60.7 days to pay for the treatment, and 66.9% of the population would be pushed below the IPL.

*Hydrocortisone was provided even though it was not appropriate at that moment.

Similarly, Médecins Sans Frontières reported an increase in mortality rate, from 0.47% to 10%, after they switched to another antivenom for six months due to unavailability of FAV-Afrique [33]. Because of this, patients lose trust in the snakebite care offered, leading to delays in seeking care and increased use of traditional treatments by victims, and loss of trust in antivenom efficacy by healthcare workers [11,32]. In Ghana, when the snakebite mortality rate in health facilities dropped again, a 50% increase in snakebite patients' attendance was observed [32].

Depending on the type of antivenom administered, several severe complications can arise, including anaphylactic reactions, occurring in up to 40% of patients, and serum sickness, occurring five to 14 days after antivenom administration [34–36]. Adrenaline is the recommended prevention and treatment method for anaphylactic reactions, while chlorpheniramine and prednisolone are used for mild and severe cases of serum sickness, respectively [34–36]. In neurotoxic envenomings there is also a risk of respiratory failure, which is managed through ventilatory support, consisting of endotracheal intubation or assisted ventilation, and in the case of neurotoxic cobra bites also with neostigmine and atropine [4,35,37]. We have shown, however, that availability of these commodities for managing complications is very low across facilities in Kenya, a situation exacerbated by stock-outs. Considering these adverse reactions are common in envenomings and can lead to death if not managed, ensuring the availability of both antivenom and associated commodities at facilities for quick access is critical. To facilitate this, snakebite should be made a part of routine national surveillance, with mandatory recording of the number of snakebites admitted to health facilities at all levels. Second, more data is needed on the effectiveness of the available antivenoms in Kenya. A case reporting system should therefore be established, in which snakebite cases presenting to health facilities and the subsequent care provided are recorded. This system should include the reporting of the type of

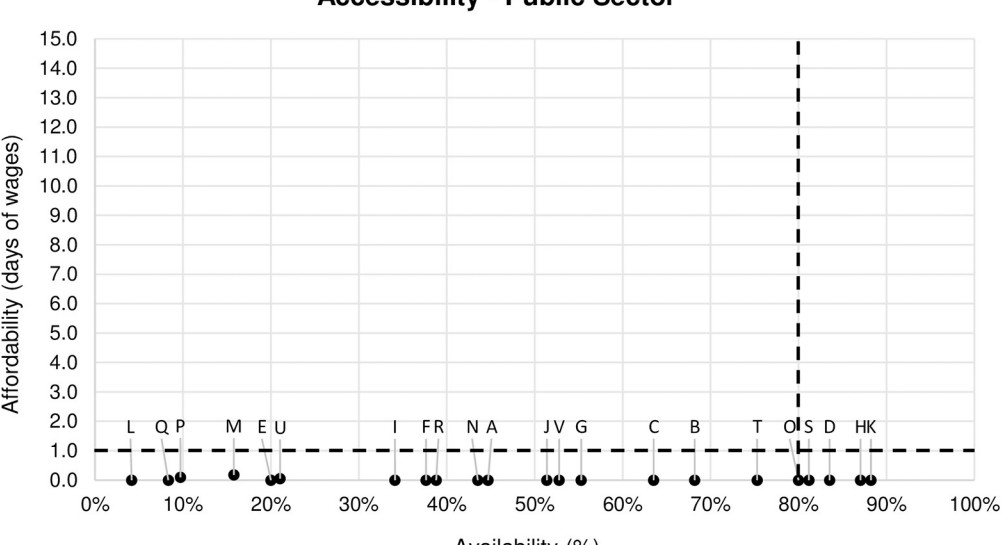

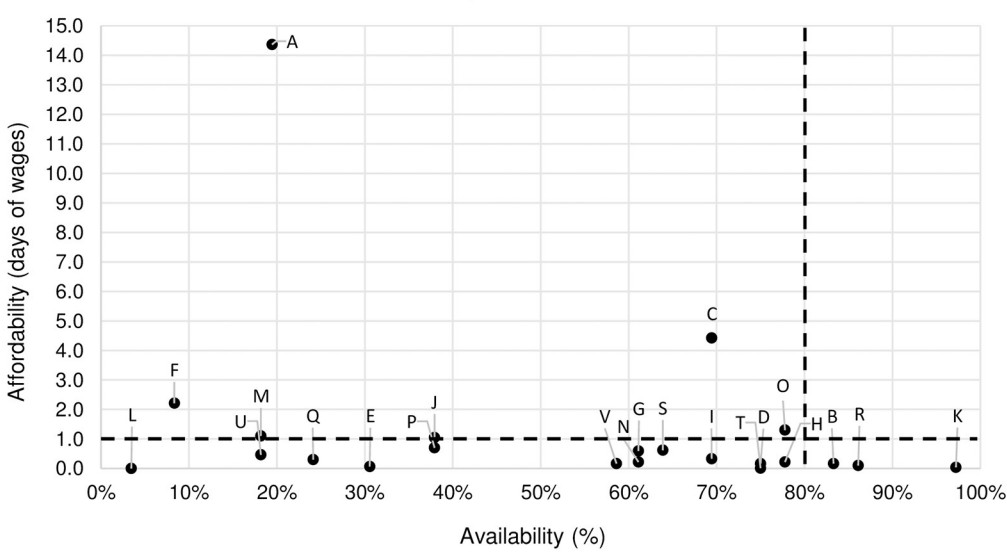

[a]A: antivenom; B: tetanus vaccine; C: benzylpenicillin; D: metronidazole (200mg or 400mg); E: metronidazole (200mg/5ml); F: gentamicin (10mg/2ml or 20mg/2ml); G: gentamicin (40mg/2ml or 80mg/2ml); H: amoxicillin (250mg); I: amoxicillin (500mg); J: amoxicillin + clavulanic acid; K: paracetamol; L: dihydrocodeine phosphate; M: morphine; N: adrenaline; O: hydrocortisone; P: chlorpheniramine (10mg/1ml); Q: chlorpheniramine (2mg/5ml); R: prednisolone; S: Saline; T: lidocaine; U: neostigmine; V: atropine.

**Fig 1. Accessibility of snakebite commodities, per sector.**

antivenom administered, other commodities used, and the health outcomes of the patients, including any adverse reactions. This would allow policymakers to map facilities with a high case rate and respond by stocking suitable antivenom and other supportive treatments.

Even if antivenom and supportive commodities to manage adverse reactions are adequately available, the question remains whether healthcare workers have the skills to properly manage the patient. Research in Kenya has shown, for example, that only 12.4% of healthcare workers

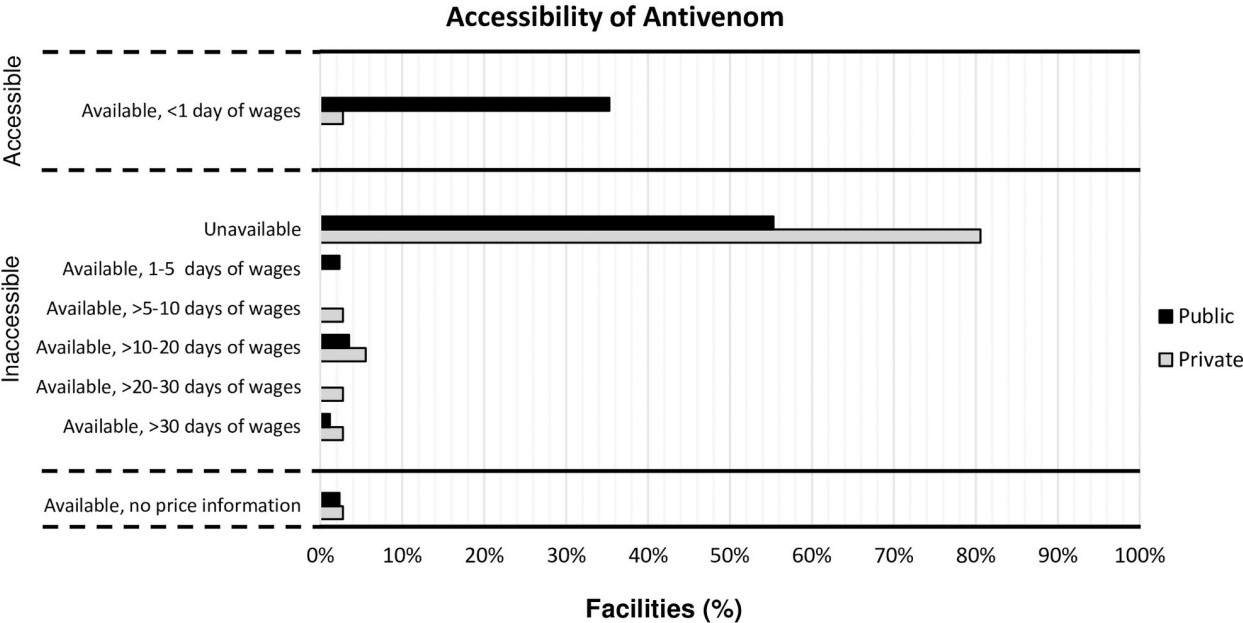

**Fig 2. Accessibility of antivenom, per sector.**

had received training on snakebite management [12]. In line with this, we found that the 20-minute whole blood clotting test (20WBCT), which is a simple test using a glass tube to test for coagulability to identify hemotoxic envenomings, was indicated to be available at only 7.5% of Kenyan facilities [4]. The question here, however, is whether the availability was actually so low, or if it was indicated to be unavailable because the healthcare workers were unfamiliar with this test, which requires only a glass tube. Efforts should therefore not only focus on improving availability of snakebite commodities, but also on increasing healthcare worker knowledge on snakebite management.

Antivenom affordability was not shown to be generally problematic in the public sector. However, in the private sector, where patients might have to buy antivenom if it is not available in the public sector, affordability can be problematic. The cost of one vial of antivenom would already impoverish 39.0% of the population if they required treatment. Considering that the average dose for seven antivenoms on the market in 2011 in sub-Saharan Africa, based on the manufacturers' recommendations, was 4.5 vials, which can go up to as many as 12 vials depending on the manufacturer and response of the patient to treatment, antivenom becomes unaffordable for almost the entire Kenyan population [7]. This catastrophic health expenditure that might be incurred by snakebite victims is also illustrated by the case example, which highlights that incurred costs are not only due to the purchasing of antivenom but are also an accumulation of the costs of treating the symptoms and complications. A study conducted in Kenya underscores the impact snakebite can have: 46% of the snakebite victims in the study noted they were unable to afford the hospital bills for their snakebite treatment, and 20% also noted they went into debt because of it [38]. Of note is that both approaches used here to calculate affordability provide only an indication of what the affordability of a commodity is. In the case of a snakebite, costs incurred are acute and of short duration, but often present an immediate financial pressure to the victim and their family. Affordability calculations like this do not take into account that patients might need to sell their valuables, livestock or land to pay for the treatment, which has long-lasting financial consequences not assessed here [39,40]. To

fully understand the socio-economic burden of snakebite on victims and their families, future research should specifically study all components of treatment affordability.

To tackle the unaffordability of snakebite treatment the cost of antivenom should be evaluated by what constitutes an effective dose rather than by the number of vials. Procurement agencies are sometimes misled by the cost per vial rather than the entire cost of effective treatment. To prevent higher total costs for an effective treatment, recommended dosages should be backed up by independent, evidence-based studies and real-world data measuring product efficacy with treatment outcomes [7]. Further, the Ministry of Health should focus efforts on ensuring antivenom is available for free to the patient at public facilities to avoid catastrophic health expenditure otherwise incurred in the private sector. The roll-out of Universal Health Coverage (UHC) in Kenya provides the perfect opportunity for improving affordability of antivenom. Including antivenom and commodities for supportive treatment and complications management in the UHC benefits package could greatly reduce the impact purchasing commodities has on a family's financial situation. Efforts should therefore focus on advocating for inclusion of these commodities, especially antivenom, in this package.

This research showed that stock-outs seemed to have worsened during COVID-19, with facilities reporting stock-outs significantly more often in the period after COVID-19 measures were implemented. These results seem to confirm stories in the media that COVID-19 has disrupted supply chains, and are in line with findings from studies on the impact of COVID-19 on the availability of antiretrovirals, which reported low levels of stock or delays in deliveries [41,42]. Health system strengthening is needed to ensure that in future emergencies or pandemics the supply of commodities is not hampered or de-prioritised, and those in need are still able to access the care they need. Further, the WHO will pilot an antivenom stockpiling programme in sub-Saharan Africa as a way to ensure access to effective antivenom treatments, which might solve some of the issues related to availability and stock-outs. At the same time, questions related to the sustainability of such an approach have been raised by Habib et al (2020) [6].

## Strengths and limitations

This research is the first to study the availability, stock-outs and affordability of not only antivenom, but also commodities used for supportive treatment when managing snakebites. A standardised and validated methodology was used [20]. Nevertheless, this research also knows some limitations. Due to the non-probability sampling and the distribution of public, private and PNFP facilities within the counties, we only surveyed a limited number of private not-for-profit facilities, which made it impossible to analyse this sector separately. Furthermore, the WHO/HAI methodology measures availability of commodities at one point in time. To mitigate this, we included commodity stock information for a period of twelve month to provide an indication of what the availability might be throughout the year. However, because availability of some of the commodities was very low, stock information could not be analysed for all the commodities as the commodity was never stocked at that facility. The WHO/HAI methodology further calculates affordability using the wage of a lowest-paid government worker. However, as seen in this study, the wage of a lowest-paid government worker was KSH 452.40 per day, while 24.2% of the population was living below the poverty line of USD 1.90, which was equal to KSH 202.58. The wage of a LPGW is thus not a sufficient benchmark for affordability in Kenya. Anticipating this, we also used the impoverishment approach, which provides a better indication of the actual affordability for the Kenyan population. However, as described by Niëns et al., this affordability measure also provides merely an indication due to the assumptions inherent to the HHFCE calculations and the linearity of the income distribution

between groups [43]. Further, the impoverishment approach is often used for calculating affordability of medicines for chronic conditions, making it easier to calculate daily costs of a medicine. Since snakebite treatment costs are not chronic and instead incurred over a very short time period, we used the HHFCE per month to calculate affordability.

## Conclusion

This study has shown that access to antivenom and supportive treatment to manage snakebites is problematic in Kenya. Availability was low, and while affordability was not a problem in the public sector, stock-outs of commodities force patients to buy them from the private sector, where antivenom in particular was unaffordable to many. Stock-outs seemed to have worsened during COVID-19, highlighting the needed for a strengthened health system that can secure continuity of care during emergencies or pandemics. To improve availability and reduce stock-outs, snakebite should be made a part of routine national surveillance, with mandatory recording of the number of snakebites admitted to health facilities to allow policymakers to map facilities with a high case rate and respond by stocking suitable antivenom and other supportive treatments. Further, in hotspot areas, a reporting system should be set up, in which snakebite cases, the provided care, and the treatment and patient outcomes are recorded and reported. Inclusion of antivenom into the UHC packages being rolled out in Kenya would further facilitate accessibility.

## Supporting information

**S1 Table. Surveyed snakebite commodities.**
(DOCX)

**S2 Table. Antivenom brands stocked at health facilities, per sector.**
(DOCX)

**S3 Table. Accessibility of snakebite commodities, per sector.**
(DOCX)

## Acknowledgments

The authors would like to thank members of the Global Snakebite Initiative for their review of the commodities list, the data collectors for their support in this research, and all healthcare workers for the time taken to participate in this study.

## Author Contributions

**Conceptualization:** Gaby Isabelle Ooms, Janneke van Oirschot, Benjamin Waldmann, Tim Reed.

**Formal analysis:** Gaby Isabelle Ooms.

**Investigation:** Dorothy Okemo.

**Methodology:** Gaby Isabelle Ooms, Janneke van Oirschot, Dorothy Okemo, Benjamin Waldmann, Eugene Erulu, Aukje K Mantel-Teeuwisse, Hendrika A van den Ham.

**Project administration:** Gaby Isabelle Ooms.

**Validation:** Janneke van Oirschot.

**Visualization:** Gaby Isabelle Ooms, Janneke van Oirschot.

**Writing – original draft:** Gaby Isabelle Ooms.

**Writing – review & editing:** Janneke van Oirschot, Dorothy Okemo, Benjamin Waldmann, Eugene Erulu, Aukje K Mantel-Teeuwisse, Hendrika A van den Ham, Tim Reed.

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
