## [Decision Letter · Decision Letter 0]

14 Jun 2021

Dear Dr Ooms,

Thank you very much for submitting your manuscript "Availability, affordability and stock-outs of commodities for the treatment of snakebite in Kenya" for consideration at PLOS Neglected Tropical Diseases. As with all papers reviewed by the journal, your manuscript was reviewed by members of the editorial board and by several independent reviewers. 

The three expert reviewers have all indicated your paper has substantial merit and novelty, however, many points of clarification have been raised. In light of the reviews (below this email), we would like to invite the resubmission of a significantly-revised version that takes into account the reviewers' comments. 

We cannot make any decision about publication until we have seen the revised manuscript and your response to the reviewers' comments. Your revised manuscript is also likely to be sent to reviewers for further evaluation.

Sincerely,

Stuart Robert Ainsworth

Associate Editor

José María Gutiérrez

Deputy Editor

Reviewer's Responses to Questions

**Key Review Criteria Required for Acceptance?**

**Methods**

-Are the objectives of the study clearly articulated with a clear testable hypothesis stated?

-Is the study design appropriate to address the stated objectives?

-Is the population clearly described and appropriate for the hypothesis being tested?

-Is the sample size sufficient to ensure adequate power to address the hypothesis being tested?

-Were correct statistical analysis used to support conclusions?

-Are there concerns about ethical or regulatory requirements being met?

Reviewer #1: While there have been many anecdotal reports of antivenom access challenges, the manuscript is one of the very rare examples to measure objectively access to essential products for management of snakebite envenoming, and particularly access to antivenom. 

The survey method is very clear. healthcare facilities in 6 health counties were surveyed. The sample is large enough to measure signficant differences. 

The list of surveyed items was defined by snakebite experts and was based on national and international guidelines. It reflects (to one notable exception described below) the list of commodities that are needed in a country like Kenya for effective management of snakebite envenoming.

The method to evaluate product affordability is robust and has been used in the past for other medicines. 

I would simply recommend to better describe what "availability" and "stock out" exactly mean in this study (see below).

Reviewer #2: The objective of the study was clearly stated. However, the authors need to state how they arrive at 6 survey regions. The authors adapted non-probability sampling method this could have been improved by using multi-stage sampling method initially and later probability sampling method.

24 health facilities were randomly selected, from a pool of what number of health facilities. The denominator need to be stated.

Why was community health services (level 1) excluded from the survey, they provide first aid services in rural communities and this can improve survival of snakebite victims.

Reviewer #3: Introduction section: 

Line 78 inter alia (typo)

Line 79 chage airway clearence to airway support 

Line 80 term resuscitation is not specific. Is this cardiopulmonary resuscitation or fluid resuscitation? Suggest remove it as CPR is (unfortunately) rarely available in in settings without access to ITU.

Line 114 suggest removing term 'first wave'. The epidemiology of COVID in Kenya has been quite different to Europe and the USA and I don't think it is possible to refer to distinct waves when transmission has been fairly constant in Kenya.

Methods section:

I would suggest adding an aim(s) section to the methods

Line 122 was random selection of facilities stratified to ensure adequete numbers of each type of facility? And what data source was used to create a master list of facilities to randomise from? Would this list have captured all facilities?

Line 147 - Need more detail about what the data collectors did. Did they visit every healthcare facility that was sampled? Did they check stocks and records themselves or interview staff?

Line 148 - Is the licensed healthcare worker someone employed by your study or by the facility?

Line 158 - remove term 'an analysis tool was used which included formulas to calculate...' Suggest change wording to 'Simple decriptive statistics were used to describe the availability of commodities, and results were stratified for public, private, rural and urban facilities.'

Line 166 - I don't understand what 'aggregate availability' means?

Line 185 - I don't understand 'an 'X' was used to denote...

Line 188 specify that it is 452 KSH *per day*

Line 190-191 is difficult to follow - suggest reword

**Results**

-Does the analysis presented match the analysis plan?

-Are the results clearly and completely presented?

-Are the figures (Tables, Images) of sufficient quality for clarity?

Reviewer #1: Results are clearly presented. The discussion section does focus on the main take away messages. Both Figures 1 and 2 clearly present the main findings in terms of availability and affordability.

Reviewer #2: Statistical analysis not clearly presented, the tables should contain level of significance in terms of p-value/confidence interval were applicable.

Reviewer #3: No mention of the response rate. Did every facility that was randomly selcted provide data? SHould report this at beggining.

Line 230 - further detail about antivenom would be helpful. Can you provide a breakdown by facility type. I would not expect dispencaries to stock antivenom as they usually do not have the facilities to administer intravenous drugs, and certainly would not be a safe place to manage an anaphylactic reaction. I appreciate you state that they are expected to stock it, but it may be worth presenting stock availability in hospitals too. Could you also present the percentage of facilities that stock antivenom but do not stock adrenaline - as all facilities giving antivenom should be able to manage anaphylaxis.

Table 2 and 3 - it is quite challenging for the reader to scan the results and see where availability is particularly low. I wonder if you could shade the cells with a gradient to depict different levels of availability. For example, see heatmap Figure 1a of Ainsworth et al https://doi.org/10.1371/ journal.pntd.0008579

Table 4 - I was surprised to see stock-outs were as low as 20% for antivenom. And only 13 days of the year was antivenom not available. To me that suggests that those facilities that do have it have fairly consistent supply. Whereas 65% of facilities don't have it at all. This may be worth touching on in the discussion.

Box 1 - Don't refer to black stone as first aid as this may suggest it is appropriate first aid. Perhaps 'traditional therapy.' Don't refer to use of hydrocortisone in first facility - this is innapropriate and shouldn't have been given. Highlight that after receiving antivenom the child developed an anaphylactic reaction that required treatment with adrenaline, hydrocortisone and chlorpheniramine. During the hospital stay the child developed a bite site soft tissue infection and sepsis that prompted administration of co-amoxiclav and gentamicin (note co-amoxiclav is given three times daily but genetamicin is only once daily). Paracetamol is usually given four times per day. 

Figure 1 - it is challnging to refer to the key for each dot on the plot. Could you use a cross table with affordability against availability. Consider heatmap as for Table 2/3. 

Discussion:

Line 405 - 'envenoming tests' is vague term. Suggest remove this sentence as the only test available in sub-Sahraan Africa is 20MWBCT. Venom EIAs are only routinely used/available in Australia. 

Line 420 'population' duplicated

In the discussion I would highlight that VINS is not supported by pre-clinical data and Inoserp is probably more appropriate (referring to Harrison's paper as you have done). Although I would add the caveat that no antivenom used in Kenya is supported by data from a randomised controlled trial. Nevertheless it is worrying that facilities have still not shifted away from VINS.

**Conclusions**

-Are the conclusions supported by the data presented?

-Are the limitations of analysis clearly described?

-Do the authors discuss how these data can be helpful to advance our understanding of the topic under study?

-Is public health relevance addressed?

Reviewer #1: The conclusions are supported by the data. This manuscript provides robust data at country level and a good baseline to measure progress in the coming years in Kenya.

Reviewer #2: Conclusion supported by data presented

Reviewer #3: Line 488 - typo remove 'there'

Line 492 - I dont feel notifiable is the correct term. Notifiable diseases are infectious and each case poses a risk to the wider public (such as a viral haemorrhagic fever). Agree with your assertion that data per facility should be routinely recorded to allow planning of stock. Perhaps 'routine national surveillance' would be a more appropriate term.

**Editorial and Data Presentation Modifications?**

Reviewer #1: A few points however require clarification in my opinion. 

LIST OF SURVEYED HEALTH PRODUCTS

I don’t understand exacly why tetanus immunoglobulins are listed by your survey. Neither the WHOI guidelines on snakebite envenomin in Africa (your ref number 4); not the Kenyan guidelines (your ref number 20) seem to be recommending tetanus immunoglobulins for the management of snakebite. Tetanus toxoid (vaccine) is recommended but not the immunoglobulins. It may be more reasonable to exclude them from the list. For your consideration. [That being said, it was interesting to read that tetanus immunoglobulins wasn’t available in any of the surveyed facilities!]

DEFINITIONS OF "AVAILABILITY" AND "STOCK OUT"

It is unclear to me what availability and unavailability exactly mean. Does it mean that the item was available /or stocked at the time of the survey? Maybe it would be useful to clarify. 

Similarly, it may be useful to clarify early in the text that the occurrence of stock-outs over the last 6-12 months was monitored only among facilities where the item is available. For example, neostigmine is available in only 17.1% of facilities while stock outs have been reported in 40% of facilities. From my understanding , this means that 40% of the 17.1% facilities in which neostigimine is available reported a stock out in the last 12 months. Did I get it right? In any case, it may be useful to clarify. 

DISPARITIES BETWEEN COUNTIES

Your analysis is based on the following dichotomies: private vs public and rural vs urban. I am however curious to know if there were disparities between counties. Kenya’s health system is very much decentralized, so I wouldn’t be surprised if some counties had better results than others. Maybe your study is not powered to measure that, since only 6 counties have been included. Yet, if you have meaningful data, I would like to encourage you to include them. If your data aren’t meaningful, then forget about my recommendation.

LINES 233-234

The text reads : “Availability of antivenom in endemic counties was 41.8%, availability in non-endemic counties was 19.0% (p=0.01).” Better maybe to refer to “highly endemic counties” and “less endemic counties”. 

BOX1

Maybe clarify that the boy received “tetanus toxoid vaccine”. It may be clearer for the reader than just “tetanus toxoid”.

LINE 255 AND LINE 405

I don’t really like the term “envenoming tests”. It sounds that those are venom-detecting tests, but this is not the case. Would it be possible to use another terminology ? ”tests for monitoring of snakebite envenoming” or something like that? 

LINE 418

You note that the average recommended dose is 4.5 vials which can go up to 12 vials. Where does that come from? From the product leaflets? Or from the Kenyan guidelines? Please clarify the source and link the text to a reference, if possible. 

ONLINE SUPPLEMENT 2

I don’t understand how VINS antivenom can be stocked in 66.67% of all facilities, while it is stocked in 66.67% of facilities in public sector and 100% of facilities in private sector. It should therefore be stocked in more than 66.67% of all facilities… What is the denominator? 

In addition there is a mismatch between the numbers in Supplemental table 2 and the number in the main text (lines 234-237: “When antivenom was available at facilities, the most commonly stocked brands in the public sector were Snake Venom Antiserum (African IHS) by VINS Bioproducts Ltd (68.4% of facilities), and Inoserp PAN-AFRICAN by INOSAN Biopharma (34.2% of facilities)”.

Reviewer #2: The manuscript need minor revision

Reviewer #3: -

**Summary and General Comments**

Reviewer #1: (No Response)

Reviewer #2: Major strength of this study is the fact that most of facilities studied were located in rural areas were snakebite is known to be a major public health problem.

The study also highlighted important facts regarding snakebite such as; Antivenom is not available in places where it is mostly needed and the fact that in sub-Saharan Africa victims travel long distances to reach health facilities where antivenom is available.

Reviewer #3: Overall highly relevant and important data on availability of essential stock for treating snakebite envenoming. Commendable use of standardised tools to assess affordability and a strength is that a range of health facilities have been randomly sampled. With above modifications I would suggest accepting this manuscript. A more genral comment is that I feel the word count could be trimmed to below 4000. This will increase uptake of readers

PLOS authors have the option to publish the peer review history of their article (what does this mean?). If published, this will include your full peer review and any attached files.

Reviewer #1: Yes: Julien Potet

Reviewer #2: No

Reviewer #3: Yes: Michael Abouyannis
---

## [Decision Letter · Decision Letter 1]

3 Aug 2021

Dear Dr Ooms,

We are pleased to inform you that your manuscript 'Availability, affordability and stock-outs of commodities for the treatment of snakebite in Kenya' has been provisionally accepted for publication in PLOS Neglected Tropical Diseases.

Best regards,

Stuart Robert Ainsworth

Associate Editor

José María Gutiérrez

Deputy Editor

Reviewer's Responses to Questions

**Key Review Criteria Required for Acceptance?**

**Methods**

-Are the objectives of the study clearly articulated with a clear testable hypothesis stated?

-Is the study design appropriate to address the stated objectives?

-Is the population clearly described and appropriate for the hypothesis being tested?

-Is the sample size sufficient to ensure adequate power to address the hypothesis being tested?

-Were correct statistical analysis used to support conclusions?

-Are there concerns about ethical or regulatory requirements being met?

Reviewer #1: (No Response)

Reviewer #2: This section of the manuscript had significant improvement base on previous comments.

Reviewer #3: -

**Results**

-Does the analysis presented match the analysis plan?

-Are the results clearly and completely presented?

-Are the figures (Tables, Images) of sufficient quality for clarity?

Reviewer #1: (No Response)

Reviewer #2: Results clearly presented. Quality and content of tables improved.

Reviewer #3: -

**Conclusions**

-Are the conclusions supported by the data presented?

-Are the limitations of analysis clearly described?

-Do the authors discuss how these data can be helpful to advance our understanding of the topic under study?

-Is public health relevance addressed?

Reviewer #1: (No Response)

Reviewer #2: Conclusions supported by the data presented.

Reviewer #3: -

**Editorial and Data Presentation Modifications?**

Reviewer #1: (No Response)

Reviewer #2: Accept

Reviewer #3: -

**Summary and General Comments**

Reviewer #1: All the comments I have made after review of the original manuscript have been taken into consideration by the authors. The revised manuscript is now very clear and accurate. I recomment to accept its publication.

Reviewer #2: Refer to previous review.

Reviewer #3: This study applied established methodology to describe the availability, affordability and stock levels of various commodities that are important for managing snakebite enevenoming. Lack of access to these essential commodities describes a situating that is probably common across much of sub-Saharan Africa, and provides important data for policy makers within the studied counties. Operational research of this nature is lacking in the field of snakebite, and provides an important insight into the challenges that clinicians and patients face. I feel the authors have appropiately responded to the reviewers' suggestions and I recommend accepting this manuscipt.

PLOS authors have the option to publish the peer review history of their article (what does this mean?). If published, this will include your full peer review and any attached files.

Reviewer #1: **Yes: **Julien Potet

Reviewer #2: No

Reviewer #3: **Yes: **Michael Abouyannis

---

## [Editor Report · Acceptance letter]

10 Aug 2021

Dear Ms Ooms,

We are delighted to inform you that your manuscript, "Availability, affordability and stock-outs of commodities for the treatment of snakebite in Kenya," has been formally accepted for publication in PLOS Neglected Tropical Diseases.

Best regards,

Shaden Kamhawi

co-Editor-in-Chief

Paul Brindley

co-Editor-in-Chief
